# A Review on Graphene (GN) and Graphene Oxide (GO) Based Biodegradable Polymer Composites and Their Usage as Selective Adsorbents for Heavy Metals in Water

**DOI:** 10.3390/ma16062527

**Published:** 2023-03-22

**Authors:** Lesia Sydney Mokoena, Julia Puseletso Mofokeng

**Affiliations:** Department of Chemistry, University of the Free State (QwaQwa Campus), Kestell Road, QwaQwa, Phuthaditjhaba 9866, South Africa

**Keywords:** graphene oxide, heavy metal ions, graphene, adsorption, biodegradable polymers, composites

## Abstract

Water pollution due to heavy metal ions has become a persistent and increasing problem globally. To combat this, carbonaceous materials have been explored as possible adsorbents of these metal ions from solution. The problem with using these materials on their own is that their lifespan and, therefore, usability is reduced. Hence the need to mask them and an interest in using polymers to do so is picked. This introduces an improvement into other properties as well and opens the way for more applications. This work gives a detailed review of the major carbonaceous materials, graphene and graphene oxide, outlining their origin as well as morphological studies. It also outlines the findings on their effectiveness in removing heavy metal ions from water, as well as their water absorption properties. The section further reports on graphene/polymer and graphene oxide/polymer composites previously studied and their morphological as well as thermal properties. Then the work done in the absorption and adsorption capabilities of these composites is explored, thereby contrasting the two materials. This enables us to choose the optimal material for the desired outcome of advancing further in the utilization of carbonaceous material-based polymer composites to remove heavy metal ions from water.

## 1. Introduction

Having access to a clean water supply is a basic need in South Africa, as well as the rest of the world. Water is one of the most crucial elements in the sustenance of life on Earth, and as such, its preservation is important. There is a growing concern regarding the availability of usable water in South Africa, as well as on a global scale. In 2007, the World Health Organisation (WHO) issued a report stating that around 1.1 billion people did not have access to a supply of clean water globally [1]. Currently, the number has since doubled to an estimated 2.2 billion lacking access to clean water on a global scale, and South Africa is amongst the most water-scarce countries [2]. Further estimates show that South Africa will have reached a physical water scarcity by 2025 and is expected to be 17% in deficit of water by 2030 [3]. All these estimates and statistics are important in understanding and thus remedying this imminent problem, as a clean water supply is crucial for human life. It is thus very important to understand the causes in this regard in order to develop an effective and systematic way to preserve and maybe harness clean water sources.

The main reasons for water scarcity in South Africa are climate change, an increasing population, irresponsible water usage, and poor wastewater treatment methods. Climate change causes global warming, which in turn results in delayed rainfall that brings about draught and a deficit in the water supply [4]. Then when it does rain, the rain does not become sufficient to entirely fill the dams, or it is overly raining such that there is flooding and water that is difficult to preserve and purify. This occurs whilst the human population itself is exponentially growing, which by logic would suffice to say the water will be used up ultimately. According to the Institute for Security Studies (ISS), each South African utilizes around 234 L of water on a daily basis, whereas the global average is 173 L [5,6,7]. This is an excess of 61 L per person and around 3.6 billion L for the whole population of South Africa daily. On top of all that, the wastewater treatment processes have not reached their full potential yet, in the country [5,6,7]. Reports show that billions of liters of sewage (untreated or poorly treated) and wastewater (pharmaceutical and industrial) are found disposed into rivers and oceans [8]. Around 56% of the country’s treatment plants are either in poor condition or the processes used to purify the water have certain drawbacks such as financial implications, reusability problems, fouling and maintenance difficulties as a result of outdated structures [8]. In the industries, it is alleged that the groundwater is greatly underused, thereby still leading to the contamination and ultimate depletion of clean water sources. There is a growing need to explore ways of effectively treating wastewater, focusing on specific contaminants, in order to assure reusability through recycling.

Amongst the different contaminants found in water are heavy metals. By definition, these are metals that have a specific density of 5 g/cm^3^ and more [9]. Examples are chromium, zinc, copper, cadmium, lead, and nickel, amongst others, and these are the main ones responsible for the contamination of water. The main sources of heavy metal contamination of the environment include industrial, agricultural, pharmaceutical, domestic effluents, and domestic sources. These processes deposit the heavy metals into consumable forms, namely food and drinking water, and this is done through transporting agents such as wind, gravity, and surface runoff. It has been reported that the concentrations of these metals in water are very small, but the health effects are dire. Some of the health implications associated with consuming water that is contaminated with these metals are bone damage, liver dysfunction, and the disruption of the functioning of enzyme proteins that contain metal ions in the body [9].

One heavy metal of particular interest is lead, which is a greyish material that naturally occurs on the earth’s crust in very small amounts. Although lead occurs naturally in the environment, industrial activities such as manufacturing, acid metal plating, ammunition development, paints and dyes formations, ceramic and glass forming, as well as fossil fuel burning increase lead concentrations deposited on the earth’s surface. The residues from these industries contain lead ions which eventually make their way into drinking water through transporting agents such as water, wind, and gravity. Research shows that adults consume 35 to 50% of lead through drinking water, and the consumption rate of children may be over 50%. Moreover, lead is not biodegradable, and this means that it would accumulate on living cells and cause fatal damage to the liver, kidneys, reproductive system, and brain, upon consuming water that has it [10,11]. The research into the removal of lead ions from solution (water) is critical and would serve to ensure public safety and that the right to sanitation is upheld.

Throughout the years, a variety of procedures have been formulated to remove heavy metals and, specifically lead from solution. Amongst others, these methods include coagulation and precipitation, membrane filtration, ion exchange, and absorption. However, most of these methods have financial implications, and/or their execution fails for various reasons and, as such, they have not been accepted for commercial usage [12,13,14]. In the membrane filtration method, low concentrations of heavy metal ions are effectively removed, according to the literature. The downside to this method is the fact that the membranes used do not have enough or a lengthened lifetime before fouling occurs, and they eventually deteriorate [15]. The organic contaminants found in most water and wastewater masses are responsible for this fouling. They become deposited in the membrane pores, thereby leading to the failure of this method. The coagulation and precipitation method involves transferring the contaminants into solids that cannot be dissolved and removing them from the water using filtration and sedimentation processes [16,17,18]. Although this method has a high metal ion removal rate, there is no inexpensive and economically viable method to discard the residue generated, and so there are financial implications with the procedure [19]. In the ion exchange method, there is a considerable tolerance of a high regeneration, and the waste is easily discarded after usage. The hurdle with this method comes with maintenance and execution, and it is very costly to put into operation and maintain this method of heavy metal ion removal [20]. This leaves the latter method, adsorption, as the most suitable and effective method of removing heavy metals from the solution, as the literature suggests.

Adsorption refers to a process whereby gasses or liquids accumulate on a substrate surface that is in solid or liquid form [21]. The phase upon which adsorption occurs is called the adsorbent, whereas the adsorbed material is termed the adsorbate. This process has been shown to yield high adsorption capacities with the ability to remove contaminants selectively. The medium on which adsorption takes place can be liquid-liquid, liquid-gas, and solid-gas, and these are very easy to attain. Ideally and theoretically, the operational costs for this method are not at all hefty [22,23]. As a result, more and more research is focused on perfecting this method and eventually reaching an optimal state of usage. Particular interest is directed to the choice of adsorbent; there has been a number of materials tested as adsorbents of heavy metals in solution. These include zeolites, modified silica gel, natural fibers, and, most recently, carbonaceous materials [24,25,26]. Zeolites and modified silica gels have been proven to have cost implications and are hazardous to the environment. Natural fibers, on the other hand, have been shown to effectively remove heavy metals from solution in large and satisfying quantities. The problem is that the fiber has a very short lifespan, which would result in its fouling and degradation [27]. This leaves carbonaceous materials as possible adsorbents of heavy metals in water.

By definition, carbonaceous materials or carbon-based materials refer to solid materials that have carbon as a main component [28]. These materials can be in powdery, bulky, or chunky forms, but they are all non-metallic. Examples, in this case, are carbon nanotubes (CNTs), biochar (BC), activated carbon (AC), and graphene (GN) together with its derivatives [28,29]. These are all forms of carbon and consist of only carbon in their structure, with different arrangements. They are characterized by a large specific surface area, a high porosity in their structures, good thermal stability, high mechanical strength, and a controllable morphology [30,31]. All these attributes, in theory, imply that these materials can be good adsorbents of heavy metals from solution. The high surface area allows for the bulk occurrence of adsorption, whereby a large number of metal ions would be adsorbed per unit of time. The porous nature of these materials would help in the effective absorption of water, such that the water will be filtered through whilst heavy metal ions are adsorbed. Then the thermal stability and mechanical strength would work to ensure that the material can withstand high temperatures and elevated stress/strain conditions, respectively [31,32,33,34]. Lastly, the fact that the morphologies of carbonaceous materials can be controllable opens the way to a whole lot of modifications that can be done to modify the properties of the adsorbents. These can be achieved by forming composites with different materials such as polymers, metals, and inorganic substances, amongst others. Further studies show that the presence of carbon only on the backbone of these materials limits their effectiveness in the removal of heavy metal ions from solution. The literature has also illustrated that primary carbonaceous materials have very good water uptake properties, but their metal adsorption capabilities are not satisfactory [35,36]. In light of this, looking into modified versions of these structures has prevailed, with a specific interest in the derivative structures of graphene. The derivatives of graphene include graphene oxide (GO), reduced graphene oxide (rGO), fluorinated graphene, graphyne, and graphdiyne, amongst others [36]. Generally, all these derivatives have been proven to possess some metal adsorption capacities. However, it has been established that materials with oxygen (GO and rGO) exhibit higher metal adsorption capabilities. This is due to the fact that oxygen traps metal ions chemically on its surface, thereby resulting in the chemical adsorption of heavy metal ions on to the adsorbent. This happens while there is physical adsorption taking place as well due to the adsorbing structure of carbonaceous materials [37,38,39,40,41]. As rGO has a somewhat limited number of oxygen atoms, it is logical to look at GO as a possible adsorbent of heavy metal ions from solution. This work explores the work done on GN and GO, together with their polymer-based composites, as they were used to selectively adsorb heavy metals from solution.

## 2. Graphene and Graphene Oxide

### 2.1. Graphene

Graphene (GN), or fully hydrogenated graphene, is a form of carbon that has a single layer of atoms having a two-dimensional honeycomb lattice arrangement in its nanostructure. The derivation of its name is from graphite, and the suffix -ene alludes to the fact that it has numerous double bonds. Graphene possesses useful properties such as heat conduction, electricity conduction, high tensile strength, and its ability to absorb light in all wavelengths. In material science, it gained popular usage and attraction due to its high specific area, which causes ease of processability and allows for tailoring of its properties into what is required [42,43]. Figure 1 shows the structure of graphene, depicting the arrangement of carbon atoms.

#### 2.1.1. The Morphology of Graphene

Morphology is the study of form via structure, size, and shape. It gives an indication of the physical and chemical properties of nanostructured materials, making it very crucial in material studies [45]. Morphology studies are very helpful in the identification of materials through the verification of their chemical and/or physical structures. There have been quite a number of these studies done on graphene for the purpose of in-depth knowledge of its structure and, thus, properties. In one study of graphene morphology and topography by Robaiah et al. [46], graphene was synthesized following the chemical vapor deposition method (CVD), with nickel as a substrate, using refined palm oil (RPO), refined corn oil (RCO) and waste cooking palm oil (WCPO). From there, scanning electron microscopy-energy dispersive spectroscopy (SEM-EDS) and atomic force microscopy (AFM) characterization techniques were performed to make a comparison of the yields of GO for the different oils used. SEM images showed visible hexagonal graphene growths on each nickel substrate for every type of vegetable oil used. The growths appeared flaky, and their diameter increased with an increased enhancement of the nickel substrate used. These observations were, according to the authors, an indication of the physical structure of graphene and gave insight into methods of optimizing the synthesis process. In EDS, the elemental compositions are presented in Table 1 below. From the table, it is worth noting that the lowest content of carbon and oxygen was observed when refined palm oil (RPO) was used (0.91% carbon and 15.97% oxygen), and the highest content of these two elements was observed when waste cooking palm oil was used (1.54% carbon and 21.13% oxygen). The refined corned oil (RCO) yielded averaged values of the elemental compositions (0.96% carbon and 16.09% oxygen). These results showed that WCPO allowed more diffusion of carbon atoms during synthesis and that RCO and RPO did not allow as much carbon to settle and form graphene. The attribution was said to be the molecular structures of the vegetable oils used; diffusion of carbon would occur more efficiently if the oil components are not held by very intermolecular forces. As a result, WCPO was validated as one of the precursors that yield a higher, good-quality graphene yield if nickel is used as a substrate.

In AFM analyses, it was established that the hexagonal growths are visible on the surface of the substrate. The thickness of the surface increased as the vegetable oils were allowed to interact with the substrate and form graphene. This brought about the assumption that graphene will almost always form hexagonal growths on the surface of the substrate, and this would influence the arrangement of its particles. Table 2 shows the value of average roughness (Ra) when nickel is used as a reference and different carbon sources/vegetable oils are used to this effect. From Table 2, it was further deduced that the thicker the average roughness, then the more graphene yield was formed, and it was clear that WCPO was proven to be the most optimal source for synthesizing graphene.

Other morphology studies of graphene have utilized X-ray diffraction (XRD), Transmission electron microscopy (TEM), and Raman spectroscopy together with SEM characterization techniques. Monti et al. [47], in their morphology and electrical properties of graphene—epoxy nanocomposites, particularly used these techniques. XRD analyses of pristine graphene showed a peak at 26°, which corresponded to an interlayer spacing of 0.335 nm between respective graphene sheets. This interlayer spacing is similar to that of graphite and therefore validated that the graphene was not in an oxidative state. TEM and SEM micrographs indicated the presence of layers on the pristine graphene structures. These layers had a certain degree of exfoliation and were stacked together. This served as confirmation of the layered arrangement of graphene structures, although the techniques did not give an indication of the exact number of layers. The exfoliation of these layers/sheets was attributed to the sonication process involved in the synthesis of graphene. That is, the sonication process exfoliates these layers and reduces their surface area as well as their lengths, which can be useful in improving the dispersion of graphene in polymer matrices. Then Raman spectroscopy studies portrayed a 2D peak at 2700 cm^−1^ and a D peak at 1360 cm^−1^. These peaks represented the presence of more than ten layers of graphene structures and a depiction of the disordered nature of these layers, respectively. The ultra-sonicated graphene showed a second-order peak at 960 cm^−1^, whereas pristine graphene did not have this peak. This implied, according to the authors, that pristine graphene has thicker layers as compared to ultra-sonicated graphene. From this, it was concluded that the process of ultra-sonication in preparing graphene results in thinner layers of product, which might disperse easily in a polymer matrix.

In terms of thermal properties, there really have not been studies that focus on the thermal stability and melting and crystallization of graphene. The current focus has been on the thermal conductivity of graphene. This might be because graphene consists only of carbon in its structure, so it might be simple to deduce its thermal stability, melting, and crystallization properties.

#### 2.1.2. Water Absorption of Graphene

Graphene has been proven to have very poor water intake abilities. This surely is because of its structure, which consists mainly of carbon and no other functional groups that may aid absorption. There really is not much to look at in the way graphene is used in water absorption studies. The consensus here is that the introduction of graphene to any material will lower the water intake properties [48,49]. This is easily attributed to the said structure of graphene, which does not have functional groups that are hydrophilic.

#### 2.1.3. Graphene as an Adsorbent for Heavy Metals in Water

It has already been established that graphene does not possess good water intake properties. As such, researchers shy away from using it as an adsorptive material for heavy metals in aqueous solutions. The already reviewed structure of graphene, which shows it to consist of only carbon, also contributes to its inability to adsorb heavy metals from water. Due to these already established parameters, it would only make sense that there is no progressive and tangible work so far attributed to the removal of heavy metals from water by graphene. There are some reviews and little work done in this regard on the feasibility of graphene being used as a selective adsorbent of heavy metals in water [50,51,52]. Most of these reviews make a comparison among graphene, graphene oxide, and multi-walled carbon nanotubes (MWCNTs). The outcome is more or less the same: The implied effectiveness in the removal of heavy metals decreases as follows: GO > G > MWCNT. This solely means that graphene oxide would make a more effective adsorbent than any other carbon-based material, whether used individually or as a composite in polymer matrices. It is also posited that graphene becomes a better adsorbent of organic pollutants such as dyes and phenolic compounds in water, as was reported by Ramirez et al. [51] when they reviewed graphene materials for removing organic pollutants and heavy metals from water. This is said to be due to the chemical interaction that takes place between the graphene surface and the pollutants. The large surface area of graphene’s surface, with pi-electrons, forms strong bonds with organic pollutants such as dyes. However, the maximum adsorption efficiency that can be reached by graphene and its composites is around 10%, whilst graphene oxide can go all the way to 90% [50,51]. It has also been suggested that the development of porous graphene and other carbonaceous would aid in its hydrophilicity and, thus, adsorption efficiency. This is because sometimes the pores act as absorption initiators and adsorption sites, resulting in the passing of water and trapping of adsorbates on the surface of the adsorbents [50,51,52]. To this effect, it suffices to posit that graphene cannot be relied on entirely to remove heavy metals such as lead, zinc, and copper, from the solution. This would introduce graphene oxide as a candidate to be reviewed as well.

The current work on graphene on its own has been remarkable and progressive; most researchers have contributed profusely to the general understanding of the structure of graphene. With more understanding, it becomes eminent that the water uptake and metal ion intake properties are very weak for graphene. On top of this, it is considerably costly to purchase or manufacture. Hence not many researchers have opted to utilize it. All research points towards graphene oxide in this regard, as it seems to be a promising substitute.

### 2.2. Graphene Oxide

The functionalization and exfoliation of graphite and graphene forms graphene oxide (GO). This is a carbonaceous carbon allotrope consisting of a single graphite oxide layer. The synthesis of GO is done by chemically treating graphite in an oxidation reaction while continuously exfoliating or dispersing it in a solvent. To this day, the precise structure of GO has not been described conclusively. However, a series of models have surfaced that aim to describe its structure. The common aspect of all these proposed models is that they all suggest that GO consists of oxygen-containing functional groups of variety, with specific areas of attachment on the structure [53,54,55]. In one study by Kim et al. [56], the authors analyzed the structure of graphene oxide in order to identify the oxygen-containing functional groups and their position on the GO structure. From this, it followed that firstly there are ester and hydroxyl groups. These groups were identified to be situated on the basal plane of the GO structure. Then follows the carboxy, phenol, carbonyl, quinone, and lactone groups, which are found in small quantities and are situated on the edges of the GO sheets. This does not take away from the fact that the precise structure is still uncertain, and the primary contributing factors are difficulty identifying the oxygen-containing functional groups with their positions and distribution on the GO sheets, lack of completely successful methods to synthesize it, and the fact that the quantities of atoms making up GO are not quantitatively proportional [57]. So there is a lot more to discover about GO in terms of its structure, properties, and applications.

#### 2.2.1. The Morphology of Graphene Oxide

Several morphology studies have been conducted on GO in order to verify its structure and/or synthesis. Shang et al. [58] looked into the synthesis and structure characterization of GO. For their morphology analysis, the authors used atomic force spectroscopy (AFM), X-ray photoelectron spectroscopy (XPS), and Fourier transform infrared spectroscopy (FTIR). Their findings for all three techniques are illustrated in Figure 2 below.

From the results above, the AFM images showed the average lateral sizes or thickness of GO sheets to be 1 µm and a height of 0.919 nm. This was indicative of the monolayer structure of the GO sheets, according to the authors. The XPS analysis resulted in a C1 spectrum having four different peaks. These were: C=C (284.3 eV), C-O (286.3 eV), C=O (287.6 eV), and COOH (289.0 eV). The peaks were a portrayal of the existence of oxygen-containing functional groups in the structure of GO. Then in FTIR analysis, a spectrum was obtained with the following peaks and their functional groups: C-O (alkoxy) at 1045 cm^−1^, C-O-C (ester) at 1226 cm^−1^, and C=O in carboxylic acid and carbonyl moieties (carbonyl) at 1719 cm^−1^. There were also bands observed at 3330 and 1396 cm^−1^, which were attributed to the O-H stretching mode and deformation vibration of intercalated water, respectively. The authors stated that GO was successfully synthesized in light of these observations. Furthermore, they pointed out that the observed FTIR and XPS results implied that GO is amphiphilic, has a hydrophobic basal plane, and is hydrophilic at the edges. In another study by Ghosh et al. [59], the morphology and property analysis of GO was performed. The authors used FTIR, Raman spectroscopy, and X-ray diffraction (XRD). Their FTIR analyses were more or less similar to those obtained by Shang et al. [58], where there was a broad peak at 3441 cm^−1^ which was attributed to the stretching of the O-H bond and portrayed the presence of a number of OH groups on the surface of GO. A band occurred at 1760 cm^−1^, indicating the C=O stretching, and another at 1403 cm^−1^, which represented the presence of carboxy groups in GO. The Raman spectroscopy and XRD results also validated the successful synthesis of GO. In Raman analysis, a spectrum was obtained with two main features: The G band and the D band. The G band was said to represent the first-order scattering of phonons by sp^2^ carbon atoms, while the D band was representative of the breathing mode of photons. The G band was found at 1591 cm^−1^ and was a broader band compared to the D band, which was found at 1341 cm^−1^ and was more apparent. The extensive oxidation of graphite to graphene oxide caused a distraction of the sp^2^ character and formed defects in the GO sheets, thereby resulting in the said shifts in bands. Then, their XRD results showed a diffraction peak at 11.41°, with a calculated interlayer spacing of 0.741 nm. This, according to the authors, was indicative of the intercalation of water molecules into the graphite layers and the formation of oxygen-containing functional groups between the layers of graphite. Several other authors arrived at similar findings when analyzing the morphology of GO [60,61,62,63]. A typical structure of GO is shown in Figure 3. It clearly shows the presence of oxygen-containing functional groups on GO.

#### 2.2.2. Thermal Properties of Graphene Oxide

Thermal properties refer to the physical properties of a material that are related to heat conductivity. Briefly put, these are properties that a material exhibits when put under a fluctuating heat cycle. Four major components of thermal analysis have been established, namely: heat capacity, thermal expansion, thermal conductivity, and thermal stress [64]. A very limited amount of work has been published on the thermal properties of GO [65,66]. Alhassan et al. [65] performed differential scanning calorimetry (DSC) and thermogravimetric analysis (TGA) on GO. On their DSC results, two distinct peaks were observed at 135 °C and 185 °C, respectively. The first peak was endothermic and attributed to the detachment of water molecules from the surface of GO. Then the second was an exothermic peak, which was reported to represent the thermal decomposition of oxygen containing functional groups on the GO. The transition at 135 °C was also seen in TGA as a mass loss, thereby validating the said detachment of water molecules. Furthermore, the TGA curves showed a mass loss at 200 °C, which was attributed to the evolution of carbon dioxide and carbon monoxide from the molecule. Then Bhawal et al. [66] synthesized and characterized GO, focusing specifically on the thermal degradation properties of GO using TGA under a nitrogen atmosphere. This TGA results showed a two-step degradation process, with the first step being below 100 °C and the second occurring between 176 and 201 °C. The first degradation step was attributed to the detachment of water molecules and also validated the hydrophilicity of GO. Then the second step represented the decomposition of oxygen containing functional groups from the structure of GO. From there, the main decomposition of carbon occurred. Although there were no reports of char content being left after thermal decomposition by the authors, their TGA curve for GO clearly showed that char content of around 10% remained. This is common with most carbonaceous materials when the TGA is done in a nitrogen atmosphere. The nitrogen acts as an inert gas, thereby preventing the oxidation reaction of the analyzed material. This then results in the formation of char, as most of these materials possess aromatic rings, which remain residuals that are hard to decompose after the TGA run. The thermal stability of GO has been proven to be low, and this would disadvantage its usage in applications that subject it to high heat conditions.

#### 2.2.3. Water Absorption Capabilities of Graphene Oxide

The area of study that looks at the water intake properties of GO, according to my knowledge, is still very novel, and as such, there have been minimal outputs in this field [67,68]. As already established, the water absorption capabilities of graphene are very poor, so the introduction of oxygen containing functional groups might just work to remedy this. That is, the formation of graphene oxide might result in high water intake properties, and this is where there is growing research interest. Liu et al. [67] investigated the effect of humidity on the water absorption capabilities of GO. The authors found out that the water absorption capacity of GO rises with an increase in the mass of GO powder used in grams. These general observations were attributed to the oxygen-containing functional groups in GO. These groups were said to act as sites of intercalation that allow the water to be diffused through while the solid material is adsorbed onto the surface of GO. As the humidity was increased, the absorption equilibrium was reached faster, which was attributed to the fact that GO is hydrophilic in nature. In another study by Lian et al. [68], the water intake kinetics of GO were explored. The authors observed that the maximum absorption capabilities of GO reached 0.58 g water per gram of GO. This is a high intake capacity portrayed by GO and ascertains the assumption that it has high water absorption capabilities. The authors also compared the water intake properties between flaky GO and ground GO. The grounded GO reached equilibrium at a faster rate than that which was flaky. The reduced particle size in ground GO allowed ease of penetration of water, according to the authors. The ground GO, however, showed lower absorption capacity. This observation was because, with the grinding, the capillary transportation of water in GO is disrupted, resulting in reduced durability and, thus, absorption capability.

#### 2.2.4. Graphene Oxide as an Adsorbent for Heavy Metals in Water

Although there has not been much work published [69,70] that looks at the heavy metal ion removal of GO, this field grows each day as it is promising. Sitko et al. [69] conducted a study on the effectiveness of GO in removing Pb(II), Cu(II), Cd(II), and Zn(II) ions from water. The authors investigated the influence of pH on the number of heavy metals adsorbed, as well as on the concentration of heavy metals adsorbed, by GO. Their results for the effect of pH on the adsorption capacity are represented in Figure 4 below. The authors quickly realized and stated that the oxygen containing functional groups were the primary causes of adsorption. They went on to establish that the Pb(II) ion had been adsorbed the most by GO, as compared to the other heavy metal ions. This was done at the optimal pH range of 4–8 as all the samples were analyzed.

Following this, the authors discovered that adsorption had proceeded by chemisorption for the most part of the experiment. Chemisorption is a type of adsorption occurring on a chemical basis, with a possibility of an irreversible reaction between the adsorbent and the adsorbate. Then they did kinetic modeling to illustrate the mechanism involved in GO adsorbing these heavy metal ions. The Langmuir and Freundlich models were used, and the following Table 3 was obtained.

Where qmax is the maximum amount of metal ions adsorbed per unit mass of GO at high equilibrium ion concentration (mg·g^−1^), K_L_ represents the enthalpy of adsorption (L·mg^−1^), K_F_ and n are the Freundelich constants related to the adsorption capacity and adsorption intensity, respectively. The qmax values obtained for all the metal ions implied that 1 g of GO could adsorb around 5 mmol of divalent metal ions. This makes GO the highest adsorbing material currently, amongst all the other materials, according to the authors. Moreover, the constants obtained on the Freundlich model were between 1 and 10, and this implied that adsorption was favored under the studied conditions. Upon fitting the isotherms from data derived from the table above, the authors obtained Langmuir and Freundelich plots shown in Figure 5. It was established that a more linear fit was obtained for the Langmuir model. As such, it was stipulated that the adsorption of the heavy metals listed in the table followed the Langmuir model. This means that adsorption took place on a monolayer basis, in which heavy metals were adsorbed into generally one homogenous surface instead of a heterogeneous one.

The general trend in observations is similar for these researchers. For instance, Zhao et al. [70] looked at the removal of Pb (II) ions from solution using a few layered graphene oxides, and also posited that adsorption depended on the oxygen containing functional groups. The authors went on to analyze the adsorption in terms of the pH of the medium, ionic strength, and concentration. The observations in this regard implied that the process of adsorption was more reliant on the pH than it was on the other mentioned aspects. This followed the observations that the adsorption of Pb(II) ions steadily increased at the pH range of 1 to 8. Then increasing the pH further resulted in an adsorption capacity reduction. That is, it was established that more Pb(II) ions were adsorbed in acidic media than when the media was basic. It was then suggested that in basic media, the hydroxide ions develop into large and complex structures with the metal ions, resulting in difficulty of adsorbing the metal ions. In modeling their results, the authors deployed Langmuir isotherms to assess the adsorption of Pb(II) ions from the solution. From this, the maximum adsorption values of 842, 1150, and 1850 milligrams per gram (mg/g) were obtained at temperatures 293, 313, and 333 K, respectively. In the Freundelich isotherm model, constants were obtained, which were all less than 10. The conclusions drawn from these observations were that adsorption took place on a homogenous surface for the better part of the analysis. After this, the authors contrasted the different maximum adsorption capabilities of different adsorbents. These adsorbents were sawdust, activated carbon GMZ bentonite, iron oxide, oxidized multiwalled carbon nanotubes, hazelnut shells, graphene, and graphene oxide. The analyses were done at a pH of 6 and at room temperature (298 K). The results that followed showed, as was expected, that GO had the maximum adsorption, with a value of 842 mg/g of Pb(II) ions adsorbed. This adsorption was a 50% marginal difference from the other adsorbents, and this validated the idea that GO possesses optimal metal adsorption capabilities. It is clear that GO is a very hydrophilic material, capable of absorbing large volumes of water and adsorbing metal ions at an effective rate.

GN and GO have received limited attention as adsorbents of heavy metals in water, which is owed to their difficulty in dispersing, which leads to agglomeration and shortened life span. With this arises the need to musk the adsorbents in order to prolong their usage and optimize it, and polymers have seemed the ideal candidates for masking these adsorbents. Theoretically, forming polymer blends with GN and GO and the appropriate selection of polymers would result in materials that have improved mechanical, thermal, and adsorption properties. Further optimization would arise from utilizing biodegradable polymers, which would ensure that the adsorption process does not harm the environment in any way. Briefly, these are polymers that disintegrate due to the action of enzymes or living organisms such as bacteria and fungi, upon being discarded. These polymers contribute a great deal to green chemistry and the conservation of the environment. Some examples of biodegradable polymers are poly(lactic acid) (PLA), poly (ε-caprolactone) (PCL), poly(3-hydroxybutyrate-co-3-hydroxyvalerate) (PHBV), and polybutylene succinate (PBS), amongst others. From this, it follows that the recent strides in polymeric composites of GN and GO are reviewed.

## 3. Graphene and Graphene Oxide-Based Polymer Composites

### 3.1. Graphene/Polymer Composites

Graphene/polymer composites have attracted quite a substantial amount of research attention over the past years. The many useful applications of graphene have sparked the interest of researchers to develop new and advanced materials. Most of these composites have been developed and used for improving specific properties in polymer matrices. These include compatibility and miscibility, thermal behavior and materials, and water purification, amongst others. Most researchers have opted to investigate single polymer/graphene composites instead of binary polymer/graphene composites. This implies that the work towards using binary polymer systems is still relatively new and, therefore, quite niche.

#### 3.1.1. The Morphology of Graphene/Polymer Composites

The morphology of polymer composites with graphene as a filler has been one area of particular interest to researchers. Gao et al. [71] investigated the influence of graphene filler size on the properties of PLA/graphene composites prepared by melt mixing. Graphene flakes with 1.2 μm and 14 μm lateral sizes were used, respectively, in order to assess the effect of the considerably smaller and bigger lateral sizes of graphene on the polymer matrix of PLA. Then loadings of 5, 7, 10, 13, and 15 wt.% of graphene flakes were used. For morphology analyses, SEM and XRD techniques were used, and some interesting findings were presented. In SEM, the micrographs showed more fractured surfaces and pull-outs for the larger graphene filler (14 μm lateral size) than in the smaller-sized (1.2 μm lateral size) graphene filler. The surfaces were also rougher and showed cracks where the bigger-sized filler was used. The much smaller filler composites showed a generally good dispersion even above 10 wt.% loadings, whilst the composites with a larger filler size were starting to agglomerate at loadings around 7 wt.%. These observations brought about the assumption that smaller-sized graphene particles disperse more evenly than bigger particles in a polymer matrix. It has also alluded to the fact that the affinity and adhesion of the filler depended a great deal on its size. In looking at the extent to which the filler pull-outs occurred on the polymer matrix, average particle diameters of the fillers before and after forming composites were recorded. These showed that the rate at which the particle sizes changed was more or less similar for the smaller (1.2 to 0.7 μm lateral size, 41.7% change) and larger (14 to 8 μm lateral size, 42.8% change) sized fillers. Since the smaller particles dispersed evenly, it was concluded that the larger ones were more susceptible to breaking during melt mixing. XRD measurements in this regard resulted in, firstly, an amorphous peak being observed for PLA at 16.8°. The peak was attributed to the predominant amorphous nature of neat PLA. From there, it was discovered that the intensity of this peak increased with an increase in either filler loading. This showed that the crystallinity increased as more filler was loaded, as it is known that the crystallinity is directly proportional to peak intensity in XRD analyses. The more an XRD peak is narrow with high intensity, then the higher the crystallinity of the material analyzed. At the same time, broader XRD peaks correspond to fewer crystalline sections of material. However, it is worth noting that the larger-sized filler composites showed a more intense peak in this regard. This raised the assumption that the larger the filler size, the higher the crystallinity of a material. These observations were rather odd; according to our knowledge, smaller-sized fillers are known to generally disperse evenly on the surface of a polymer matrix. This then results in improved physical interaction between the polymer and the filler, thereby increasing the crystallinity of the material. However, larger-sized fillers would not disperse evenly by forming agglomerates on the polymer matrix, resulting in poor polymer-filler interactions and a lowered crystallinity. The XRD data also showed a diffraction peak at 2θ = 26° for both composites with different-sized fillers. This was a peak attributed to the presence of graphene in the composites. However, the intensity of this peak was weaker for the composites with small-sized filler particles, and this was deemed understandable, as that particular graphene had been proven to be less crystalline. Most of these morphology studies of graphene/polymer composites indicate more or less the same. Wu et al. [72] arrived at similar findings when investigating the behavior of PLA/graphene composites. The authors investigated the morphology of composites prepared by solution mixing and characterized their samples using SEM and TEM. Images from these microscopes still showed filler pull-outs here and there, some agglomeration, but a generally good dispersion as more nanosheets were embedded into the polymer matrix. This still depicts the high affinity found between graphene and the PLA matrix. Moreover, the graphene nanosheets were observed to still be in their virgin, multi-layered structure and in stacked form. This suggested that the PLA chain is hard to intercalate into the interlayer space of the nanosheets if solution mixing is used to prepare the composites. These observations resulted in a multi-layered paper-like morphology, bringing about the assumption that the composites were, in fact, micro composites with areas of nanoparticles.

The study of the morphology of graphene/polymer composites gains more traction each day as more researchers are picking an interest in the field. The current knowledge so far gives an exciting and fruitful incite which paves the way to more advanced applications and advancements in the study of polymers filled with graphene. In time, the structure of graphene will surely be optimized and tailored for specific and useful needs.

#### 3.1.2. Thermal Properties of Graphene/Polymer Composites

Thermal analysis is very important in analyzing polymer/GN composites. This is mostly done through differential scanning calorimetry (DSC) and thermogravimetric analyses (TGA). DSC helps in understanding the melting and crystallization behavior, whereas TGA assesses the thermal degradation and stability. Very limited studies have been published that look at the thermal properties of GN/polymer composites.

##### Melting and Crystallization

The widely used characterization technique here is DSC. Generally, the literature suggests that the presence of GN in the matrix of a polymer will increase the degree of crystallinity of that polymer [71,72]. Wu et al. [72] investigated the crystallization behavior of polylactide/graphene composites with 1 wt.% GN loading. The DSC results in this regard are depicted in Table 4 below.

From the table, it is clear that both the neat PLA and the PLA/GN composite exhibited the Tg, which did not differ that much in terms of position. The second transition obtained for both samples was the cold crystallization peak. This transition shifted a bit to a higher temperature for the composite, as the table portrays. The observation was said to imply that the presence of graphene nanosheets reduces chain mobility in PLA and hinders its crystal growth. The melting transition was split into two peaks for both the neat PLA and the PLA/G composite, and its position did not change that much. So, although there were no attributions to the presence of the double melting peak for pure PLA, this could be explained by polymer science principles. That is, even though PLA did not show any crystallization peak upon cooling, it might have crystallized, but the peak was not prominent. The crystals formed from the cooling and the ones formed during cold crystallization would be different types of crystals. These two types of crystals would then melt at slightly different temperatures, thereby resulting in a double melting peak for PLA. In addition, the cold- crystallization of PLA might have resulted in the formation of two types of crystals, namely, the metastable and perfect crystals. These crystals would then melt at different temperatures, which would also result in two melting peaks for PLA. The melting transition of PLA did not change position upon GN addition, which showed that the GN did not have any significant influence on the melting process of PLA. Lastly, the degree of crystallinity of PLA was not influenced by the addition of GN. Perhaps if the analyses were done at increasing GN loadings, then there would be a visible trend in the effect of GN loading on the polymer matrix of PLA. However, on a general basis, carbonaceous materials are known to have significant effects on polymer matrices, even in small quantities. This might be the reason why the study was conducted at 1 wt.% GO content. Then Gao et al. [71] conducted DSC studies on PLA/G composites where the graphene nanoplatelets differed in particle size. The authors still established that the presence of graphene did not influence the glass transition, cold crystallization, and melting temperatures of PLA that much, regardless of their particle size. However, they also noticed that the degree of crystallinity increased as the GN loading was increased for both the small and bigger particle sizes of GN in the matrix. Murray et al. [73] investigated the thermal properties of PCL/GN composites using DSC. Their DSC heating and cooling curves are shown in Figure 6 below. The authors synthesized graphene by exfoliating graphene oxide by microwave methods (rGO) and compared it with chemically converted graphene (CCG) by forming composites with PCL for each graphene produced. The graphene loading for both composites was 1 wt.%.

The findings in this regard were that firstly, the melting temperature of PCL was not affected by the added graphene, as it remained in the range of 56–60 °C for both composites. Then during cooling, PCL, together with its composites with rGO and CCG, exhibited crystallization transitions. The crystallization temperature of PCL had a significant shift with the addition of graphene for both composites. The neat PCL polymer crystallized at 26 °C, while in the PCL/rGO composite, it crystallized at 39 °C and in the PCL/CCG at 35 °C. These observations were said to indicate that the incorporation of graphene nanosheets into the PCL matrix affects the polymer microstructure by acting as multiple nucleation centers for crystallization. In addition, in our view, graphene nanosheets might have increased the crystallinity of PCL, and the number of crystalline chains increased in the matrix of PCL.

These observations work to ascertain the assumption that the presence of graphene in a polymer matrix has a direct effect on only the degree of crystallinity. However, most research has been on one-polymer composites. Perhaps a bi-polymer composite analysis would yield different and more interesting outcomes.

##### Thermal Degradation/Stability

Thermal degradation analyses provide insight into the thermal stability of a polymeric material. This information is very important if we wish to deduce the possible applications of that material. It has been generally discovered that the presence of graphene improves the thermal stability in a polymer matrix/matrices. Rodill et al. [74] made one such observation in their analysis of graphene/polymer composites in organic media. The authors did thermal stability studies of poly(methyl methacrylate) (PMMA)/graphene composites whereby the graphene content was 1 wt.%. The observations that followed were that the onset and maximum thermal degradation temperatures of PMMA increased by 10 °C for the composites, as compared to neat PMMA. The authors also stated that the degradation rate became slower, and as such, conclusions were made that the graphene generally improved the thermal stability of PMMA. Du et al. [75] reviewed the properties and uses of graphene/polymer composites and found that the thermal stability of a polymer matrix was improved upon adding graphene to a polymer matrix. The authors established that in one study where PLA/graphene composites were investigated, the onset of thermal degradation increased by 14 °C for the composite with 1 wt.% graphene loading.

These observations by other researchers could prove useful in the study of using graphene to reinforce polymeric materials. The world needs more materials that can withstand high temperatures, and this research field promises just that.

#### 3.1.3. Water Absorption Capabilities of Graphene/Polymer Composites

Some polymers are naturally hydrophilic, and as such, researchers would fill them with graphene to assess the change, if any, in their water intake properties. Prolongo et al. [48] assessed the effect that the thickness and lateral size of graphene nanoplatelets have on the water absorption properties of epoxy/graphene nanocomposites. Different sizes of graphene nanoplatelets were filled into the epoxy matrix at 0.5 wt.% loading. Then after 15 h of immersing in water, it was discovered that the neat epoxy generally had the highest water intake, with a maximum intake of around 1.5%. All the curves started off in a linear fashion and in time reached the point of saturation. It is also worth noting that the composites which had the highest surface area (510 m·g^−2^) but the lowest average flake thickness (1.6 nm) exhibited the highest maximum water intake (around 1.5%) among the composites. This was attributed to the fact that the high surface area and less thickness in the graphene would make it more permeable and more susceptible to being porous. It is very much clear that the size and surface area of particles has a direct influence on their water intake properties. The fact that the saturation point of intake was reached after a minimal amount of time shows that graphene really does have a low water intake capability. The maximum intake of 15% for the materials used is a very low water intake percentage for both the polymer and graphene. Wang et al. [49] looked at the water intake properties of poly(vinyl alcohol) (PVA)/graphene composites. Loadings of 0.5 and 1 wt.% GO were used, and Table 5 summarises their findings.

From these results, it can clearly be seen that the absorption of pure PVA, taken after 24 h, was 105.2%, with a measured contact angle of 36°. The addition of graphene reduces the water intake and thus improves the barrier properties in the composites. This is seen by the absorption degree dropping to 59.8 and 48.8%, respectively, for 0.5 and 1 wt.% graphene loadings. The contact angle increased with the addition of graphene loading (93 and 97°, respectively). These observations clearly imply that the introduction of graphene induces and increases hydrophobicity and reduces hydrophilicity in a polymer matrix. These observations are both advantageous and disadvantageous depending on the desired objective. If the intent is to strengthen or rather reinforce the surface properties of a polymer matrix, then graphene is ideal and conducive as a filler. However, if we require to develop materials that can absorb water and probably adsorb heavy metals from solution, then graphene might just not be the go-to filler.

#### 3.1.4. Graphene/Polymer Composites as Adsorbents for Heavy Metals in Water

Unfortunately, the inefficacy of graphene in removing heavy metal ions from solution has caused a lot of researchers to refrain from exploring its polymer composites in this regard. According to our knowledge, there really is no tangible work so far attributed to the removal of heavy metal ions from solution by graphene/polymer composites. The findings presented by researchers in analysing the water intake properties of graphene-filled polymer composites just show that the inclusion of graphene yields a higher hydrophobicity. As already mentioned, it will all depend on the specific desired outcome of the prepared materials. Particularly, if we require to enhance the metal adsorption properties of polymer composites, we need to investigate fillers that have the desired functional groups, such as oxygen, for example. In light of the above-mentioned views, it becomes important that we review the properties of GO/polymer composites. The contrasts made in this regard, in comparison to graphene and other carbonaceous materials, imply that GO deserves all the attention. The oxygen-containing functional groups on its surface are the primary concern and reason to investigate the feasibility and work done so far in using GO/polymer composites as selective adsorbents of metal ions in water.

### 3.2. Graphene Oxide/Polymer Composites

The reactive sites present in GO make it ideal for structural modification and tailoring, whereby its structure can be manipulated for different purposes. This ease of processability aspect of GO has attracted the need to formulate and analyse GO/polymer composites by researchers. Although little research has been done into biodegradable polymer-based composites with GO, there has been quite a variety of papers reporting on the development and properties of GO/polymer composites in general. This has been for the purposes of adsorption of pollutants, electrical conductivity, and the improvement of materials’ properties. This section focuses on the strides made in the development and characterization of GO/polymer composites [76,77,78,79,80,81,82,83,84].

#### 3.2.1. The Morphology of GO/Polymer Composites

The limited morphology studies previously conducted on GO/polymer composites utilize SEM and TEM mostly for analysis. Generally, SEM images are not that clear at high magnifications, hence the need to supplement them with TEM, which provides clearer images at higher magnifications. The heat sensitivity of most polymers contributes to this aspect of SEM, meaning these polymers would be thermally unstable at high temperatures and result in blurred images. Pinto et al. [76] used SEM and TEM to assess the morphology of PLA/GO composites, where the GO loading was in the range of 0.2 to 1 wt.%. As expected, the SEM images were not clear, and this was because of the sensitivity of PLA towards heat, causing it to be thermally unstable at high magnifications. Then in TEM, clear GO single sheets were visible on the polymer matrix, with small aggregates here and there, as the GO loading was increased. This portrayed that the dispersion of GO in the PLA matrix was uniform, according to the authors. From SEM, spectrochemical analysis was done using Field emission spectroscopy. This resulted in the transparency of PLA/GO composites being shown to reduce with an increase in GO content. In another study by Theophile et al. [77], in their analysis of electrochemical properties of PVA/GO composites, SEM-EDS and XRD analyses were done on 50/50 *w*/*w* PVA/GO composites. Herein they observed the GO to be dispersed on the surface of the polymer matrix without being embedded deeply into the matrix. This was attributed to the smaller interlayer distances and smaller defects, thereby making it hard for the matrix to be broken down. Their EDS study showed that the carbon composition was 60%, whilst oxygen was 40% in the polymer composites. This makes sense as the GO and PVA consist mainly of carbon. In XRD studies, the peaks became more intense with an increasing GO content for a typical PVA/GO composite. This alluded to the idea that the crystallinity of the composites increased with an increase in GO content, according to the authors. Chen et al. [78] looked into the analysis of ultrahigh molecular weight polymers with GO as a filler. Loadings of GO ranged from 0.1 to 1 wt.%, and SEM was used for morphology characterization. To this effect, the addition of 0.3 wt.% resulted in an uneven fractured surface seen on SEM, with random distributions of GO in the polymer matrix. This loading (0.3 wt.%) of GO was said to result in the agglomeration and poor dispersion in the polymer matrix. However, an increase in GO content (0.5 wt.% and above) resulted in an even and smoothed-out dispersion in the matrix of the polymer, with a homogenous distribution of GO sheets. From our analysis, it is clear that GO acted as a reinforcer of the polymer system at loadings above 0.5 wt.%. GO’s affinity to the polymer matrix further strengthened as its loading was increased to 1 wt.%. The GO sheets became deeply embedded into the polymer matrix in this regard, and the GO bound more compactly into the matrix. The authors then asserted that the visible layer size was increased, which further showed and justified the affinity that GO had to the polymer and, therefore, the reinforcement effect.

#### 3.2.2. Thermal Properties of GO/Polymer Composites

Conducting thermal studies on polymer/GO composites is very important. This is done through the differential scanning calorimetry (DSC) and thermogravimetric analysis (TGA). DSC aids in understanding the melting and crystallization behavior, whereas TGA assesses the thermal stability. There has been a limited number of published work that investigates the thermal properties of GO/polymer composites [79,80,81].

##### Melting and Crystallization

Xu et al. [79] analyzed the thermal transition of poly(vinyl alcohol)/GO films using DSC with 3 wt.% GO. The resultant DSC thermograms for pure PVA and the PVA/GO composite are shown in Figure 7.

From the curves in Figure 5 above, the neat PVA had only one transition representing its melting point, at around 220 °C, with no degree of crystallinity calculated. Then for the composite, there was still one transition observed due to the melting of PVA at around 218 °C, and no degree of crystallinity was determined still. The melting point of PVA only slightly shifted (by 2 °C) to a lower temperature for the composite, which had no significant impact on the thermal transitions of PVA. The authors just put forth that the strong interfacial interactions observed on SEM could have influenced these observations. In our view, if there were strong interfacial interactions seen on SEM, there was probably a good dispersion of the filler on the polymer matrix, which would have a reinforcement effect. This would then result in an increased melting temperature or a shift to the right in the melting transition peak. The probable explanation for the slight reduction in the melting temperature observed could be due to that, although the filler interacted well with the interface, it was an inhibitor. The GO filler might have slightly prevented the GO particles from arranging in an orderly manner during preparation, thereby resulting in these crystals melting at a slightly lower temperature. To further add, the reason for the small shift could have been because the content of GO used was too small to have a noticeable effect on the melting of PVA. Unfortunately, there were no calculations of the degree of crystallinity by the authors, so the effect of GO on the crystallinity of PVA is seen. Zhou et al. [80] assessed the thermal properties of resin/GO composites with 0.5 wt.% GO loading. The authors also used PVA resin, and their DSC analyses only showed a glass transition step at around 70 °C. Then the PVA/GO composite also had one transition, the glass transition of PVA, at around 100 °C. This significant shift in the glass transition for the composite was attributed to GO delaying the transition of PVA to the rubbery state, as it increased its crystallinity. Mindivian et al. [81], in assessing the properties of PVC/GO composites, summarised their results in Table 6 below. It is clear from the table that neat PVC had two thermal transitions, namely, the glass transition at 59 °C and the melting of PVC at 302 °C. The authors then noticed that the glass transition of poly(vinyl chloride) (PVC) was reduced with the increased addition of GO.

These observations were said to imply that GO caused the molecular chains in PVC to relax with ease. Furthermore, the authors observed that the addition of 0.1 wt.% GO increased the melting enthalpy of PVC from 58.66 J/g for the neat polymer to 62.9 J/g for the composite. This was attributed to the reinforcing effect that GO had on PVC at that loading. Increasing the GO loading, however, resulted in the melting enthalpy of PVC dropping, whereby the lowest enthalpy (29.81 J/g) was when 1 wt.% GO was used. This was attributed to the weak interaction that existed between PVC and GO as the GO loading increased. The authors further stated that the formation of agglomerates of GO might have influenced this behavior. To expand further from known DSC interpretations: at low GO loadings (around 0.1 wt.%), the GO sheets might have completely covered the polymer matrix and dispersed evenly. However, when the content of GO was increased, it might have dispersed and covered all of the available surface area on the polymer matrix, such that it still remained. The remaining GO then formed agglomerates and exhibited particle-to-particle interactions. This could have been the reason for the observed agglomeration and poor dispersion, which in turn affected the melting enthalpy. The melting transition of PVC did not seem to be influenced by the addition of GO. This might imply that the GO did not have that much of an influence on the crystallinity of PVC at the loadings used. In addition, the PVC used could have been mostly amorphous.

##### Thermal Degradation/Stability

Xu et al. [79] investigated the thermal stability of PVA/GO composites using TGA with 3 wt.% GO content. For neat PVA, two degradation steps were observed at around 300 and 390 °C, respectively. Although the authors did not specify, the first step is known to be due to the escaping of side groups, namely: water, acetaldehyde, unsaturated aldehydes and ketones, benzene, and its derivatives. The water comes from the previously absorbed moisture in PVA. Whilst the other groups result from the chain scission mechanism via a six-membered ring involved in the thermal degradation of PVA. Then the second step represents the degradation of the main polymer backbone. The PVA/GO composite also exhibited two degradation steps at temperatures around 300 °C and 425 °C, respectively. The thermal stability of the polymer had improved with 3 wt.% GO addition, according to the authors. From this, it was observed that the rate at which the composites degraded was slower than that of the neat polymer. These observations, according to the authors, were an indication of the suppressed mobility of the polymer segments at the interfaces of the polymer and GO. These interactions then caused an improvement in thermal stability. Then Mindivian et al. [81] thermally and structurally characterized polyvinylchloride (PVC)/graphene oxide (GO) composites at 0.1, 0.3, 0.5, and 1 wt.% GO loading, using TGA. Their TGA analysis showed PVC to degrade in two distinct steps. The first step was at 292 °C, while the second step occurred at 452 °C. The step at 292 °C was linked to the escaping of chlorine gas, and the second degradation step represented the main chain degradation. The degradation of PVC left a char of 16%. With the addition of GO to 1 wt.%, the first degradation step moved to 276 °C, which is a decline compared to the neat PVC. This was attributed to the loss of initially absorbed moisture by GO. The second degradation step increased to 470 °C at 1 wt.% GO loading and signified that the GO improved the thermal stability of GO. The char dropped to 12% with 1 wt.% GO loading, and this was attributed to the layers in GO having a partial prevention effect on the formation of volatile aromatic compounds.

It is clear that graphene oxide-based polymer composites are still not that widely explored. Findings from different researchers in this regard are promising and can only be improved from here. GO seems to have a higher affinity towards certain polymers, and this could be very beneficial in helping to develop materials to assist in a number of problems such as pollution, drug administration, and electrical conductivity, amongst others.

#### 3.2.3. Water Absorption Capabilities of Graphene Oxide/Polymer Composites

In analyzing the water intake properties of polymer/GO composites, the generally held opinion is that the oxygen containing functional groups in most polymers and GO influence the water intake properties profusely. These groups have been proven to up the absorption rate in these materials, and as such, the minimally studied GO/polymer composites, so far, are considered very hydrophilic. In one study, Ghosh et al. [59] investigated the morphology and properties of hydroxypropyl methylcellulose (HPMC)/GO composites at 0.1 to 1.1 wt.% GO loadings. The neat polymer showed maximum water absorption rates of around 10.5%. After this, GO was loaded to up to 0.9 wt.%, and the water intake rate of the composite dropped to 9%. This observation was attributed to the formation of hydrogen bonds between GO and the polymer. These formed bonds would inhibit the interaction of water molecules with the surface of the composite. As GO loading was increased to 1.1 wt.%, the rate of water absorption increased to even surpass that of the pure polymer. The authors posited that the hydrogen bonds formed at low GO loadings were now saturated, so the excess GO had left free-running hydroxyl groups to interact with water molecules. This accounted for the increased water intake rate and justified the held opinion that GO is indeed hydrophilic, according to the authors. The other researchers, Gavin et al. [82], assessed the barrier properties of GO/vinyl ester composites using 0.5 and 5 wt.% GO contents. The authors did water intake studies based on three parameters; the water absorption rate, moisture absorption at saturation, and the coefficient of diffusion (D). The water absorption rate is the slope obtained at the initial stages of the absorption curve. Moisture absorption at saturation is the absorption when the curves are steady, and the diffusion coefficient is the amount of a particular substance that diffuses across a unit area in 1 s, under the influence of one gradient unit. The findings in this regard were still that at low GO loadings (0.5 wt.%), the rate at which the composites dropped to below that of the neat polymer. These observations were said to be due to the interaction that GO had with the polymer, as well the presence of hydrophobic material in the polymer. The calculation of the diffusion coefficient yielded similar trends, whereby small GO amounts had a lower D value, whilst, at high GO loadings, the D value went up. Conclusions drawn in this regard were that it could be that in order for the permeability of a matrix to be improved, the loading of GO should be at contents above 1 wt.%.

The above-presented work clearly shows that the water intake properties of graphene oxide and graphene oxide/polymer composites surpass those of graphene and graphene/polymer composites. The main contributing factor here would be the oxygen containing functional groups present on GO, and the assumption is that these groups would assist a great deal in metal ion adsorption. So as a step further, it would only be fitting to explore strides made in the metal adsorption capabilities of graphene oxide/polymer composites.

#### 3.2.4. GO/Polymer Composites as Adsorbents for Heavy Metals in Water

GO/polymer composites, as adsorbents for heavy metal ions in solution, have had very minimal research outputs [83,84] thus far. Most researchers have formed composites of GO with other materials such as metal oxides and sand, but not much work has been attributed to GO/polymer composites. In one study by Naushad et al. [83], biopolymer mixture-entrapped modified graphene oxide composites (1:5 *w*/*w* GO:bipolymercomposition) were used to treat heavy metal contaminated water surfaces. The authors developed the composites and investigated the effect of pH and dosage, ionic strength and associate ions, and contact time on the adsorption of Cr(III), Pb(II), and As(V) ions. In investigating the effect of pH and dosage, a pH range of 2 to 10 was used, and the results were summarised in Figure 8 below. Following this, it was discovered that around 90% adsorption of Pb(II) and As(V) took place at a pH of 3. The Cr(III) only started being adsorbed at a pH of 4, and was adsorbed the least at that initial pH range. This was attributed to the fact that the protons of the heavy metals tend to compete to react with the surface of the adsorbent, thereby hindering the adsorption process to some extent. Moreover, the optimal adsorption was observed in the pH range of 4 to 7, and this made sense as when you increase the pH, the surface becomes deprotonated to allow for adsorption. In addition, it was stipulated that the Pb(II) and Cr(III) tend to form hydroxides with hydroxide ions and become deposited on the surface of the adsorbent, thereby increasing the effectiveness of the adsorbent.

Concerning the adsorbent dosage, it was expected that increasing the dosage would result in an increased degree of adsorption. Then in investigating the effect of ionic strength and associated ions, different ionic substances were added to the adsorbent/adsorbate mixture to mimic the natural state with which the water is found if it is contaminated. The findings in this regard were that firstly Cr (III) removal was only deterred by PO_4_^3−^ ions, Pb(II) removal by increasing the concentrations of Ca^2−^, K^+^, and Mg^2+^, and lastly, the adsorption of As(V) was reduced by the introduction of the anionic species CO_3_^2−^ and PO_4_^3−^, respectively. The reasons put forth were that the removal of Cr(III) is not affected by any other ionic species but only PO_4_^3−^ due to inner sphere surface complexation at the interface of the adsorbent and metal. The reduction in Pb(II) adsorption is due to the ionic strength and charge of the Ca^2+^, K^+^, and Mg^2+^ ions being close, which might reduce the interaction between Pb(II) and the GO/polymer adsorbent. Lastly, the disturbance in As(V) metal ions by the CO_3_^2−^ and PO_4_^3−^ species was said to be because these ionic species were competing with As (V) for adsorption sites via inner-sphere surface complexation. Concerning the contact time, as expected, metal ion adsorption would increase with an increase in contact time with the adsorbent. Modeling was done in this regard, and the Langmuir isotherm proved more applicable to the scenario, with constants calculated supporting a high adsorption capability. From this, the authors tried to determine the mechanism involved in the adsorption of these heavy metals onto the prepared composites. This was done using FTIR and XPS after the adsorption of heavy metals. It was established that after adsorption, new peaks were formed in FTIR, which portrayed the presence of metal ion and oxygen bonds for all the analyzed heavy metals. This brought about the assumption that adsorption took place on the surface of the adsorbent as the oxygens interacted with the heavy metal ions. Furthermore, the XPS analyses also resulted in newly formed peaks, as well as the shifting of peaks, signifying the presence of newly formed bonds. These bonds were still said to be between the oxygen groups on the adsorbent and the adsorbates (heavy metal ions). To give it more context, the authors went on to propose that the adsorption process was mostly caused by surface complexation instead of electrostatic contact. The proposed mechanism for the adsorption is shown in Figure 9 below, clearly showing the importance of the presence of oxygen-containing functional groups on your adsorbent. From this mechanism, it can be deduced that the higher the surface area of your material, then the more oxygen-containing functional groups you can attach and increase adsorption. In addition, if porous materials are developed, then this would allow for the successful absorption of water whilst the heavy metals are trapped in the adsorption sites, thereby optimizing the adsorption process.

It was further established from the modelling that the introduction of a polymer matrix/matrices to a GO-based adsorbent would increase the adsorption capability. This is obviously due to the presence of oxygen containing functional groups on some polymers, which would act as adsorption sites to trap the metal ions. Although there has not been much work done in using these composites for heavy metal removal, the chemistry coupled with recent work really proves that GO-based polymer composites possess excellent metal adsorption properties. In one other study by Peng et al. [84], the analysis of chemically reduced graphene oxide composites with polypyrrole, chitosan, aromatic diazonium salt, and ethylene diamine was done. Although specific material compositions were not specified, it was generally observed that the different functional groups of the composites influenced the adsorption efficiency of heavy metals. Furthermore, the authors determined that greater adsorption capacities were observed when GO was mixed/masked with materials with oxygen containing functional groups. This implies that cognition should be taken when masking GO, that if optimal adsorption results are required, then the material used to mask it should have numerous oxygen containing functional groups as well. The authors also performed desorption studies and established that the GO/polypyrrole composites exhibited a 95% reusability effectiveness after four usage cycles and a 90% reusability after eight cycles. These results would pave the way for the commercial usage of these composites for the purposes of cleaning water.

The above report on GO/polymer composites is very minimal, and this is because there has not been much work done on this front, especially where biodegradable polymers are used. So, it would be fitting to identify this area as a knowledge gap and deem it only logical that the gap is explored fully and means to close it are put in place. This would contribute a great deal to the ongoing developments in material science and scientific research in general.

## 4. Conclusions and Future Perspectives

This work aimed to review and contrast graphene and graphene oxide and their polymer composites in terms of metal ion adsorption and other properties. Insight into the strides made so far in this field of research has been provided with utmost certainty. In terms of structure and morphology, recent work has established that the main difference between GO and GN is the presence of oxygen-containing functional groups on GO, whereas GN only consists of carbon. Coming to the morphology of the composites, it was established that the layered structures of both GO and GN assist with adhesion and enveloping the polymer matrix/matrices. As such, we can say that both GO and GN have a compatibilizing effect on polymer matrices. From there, the thermal properties were reviewed, starting with melting and crystallization, then thermal stability. The work done on the thermal properties in this regard is also positive and promising. In melting and crystallization, the introduction of GO and GN into a polymer matrix/matrices is said to improve the miscibility of the blend composites and increase the crystallinity. This, of course, can open the way to many uses and applications. The thermal stability is also improved with the introduction of these two fillers, according to thealready-statedd literature. Then we looked at the water absorption capabilities of GN and GO, together with their polymeric composites. Here it was established, through the literature, that the water intake properties of GN are not at all satisfactory, even when used in polymer composites. This is particularly disadvantageous. As such, GO seems to be a suitable stand-in candidate. That is, both GO on its own and in polymer composites have been proven to show great water intake properties. The idea is that the oxygen containing functional groups on GO assist a great deal with this. Then the recent work done in metal ion adsorption shows that GN and its composites are not at all suitable adsorbents of heavy metals in solution. The lack of adsorption sites is the main reason here, and as such, GO would need to step in again. Both neat GO and GO/polymer composites have been shown to have excellent metal ion adsorption capabilities. This is one area that can be further explored, as there are gaps in it. Firstly, as already stated, the GO on its own has a shorter life span, so the gap lies in the need to mask it. Then, the masking that has taken place currently mostly utilizes a single polymer. We need to look into using binary polymer systems to mask GO, and improve other properties. Lastly, a very big gap remains in the type of polymers used. There has not been much research directed to the formulation and usage of completely biodegradable polymer/GO composites for the purposes of removing heavy metal ions from solution. These polymers are important as most of them contain oxygen in their structures, which could aid a great deal in trapping metal ions from water. In addition, as most biodegradable polymers are either too brittle or too rubbery on their own, blending them would assist in remedying this and balancing out the properties to obtain optimal materials. It is very clear now which direction should be taken from here in order to advance further the knowledge already acquired to optimize the usage of GO and its composites.

As for the future perspectives regarding the usage of graphene and graphene oxide-based materials for water treatment, it is clear that graphene oxide is the most suited because of the oxygen-containing groups. The limitation with graphene is the absence of these functional groups that would act as adsorption sites and improve the adsorption rate and efficacy. Even if polymers that have these functional groups were used with graphene to form composites, the adsorption rate would surely be lower as compared to materials containing graphene oxide. Therefore, as it stands, there are many countries trying to explore the usage of graphene oxide-based materials in water treatment applications. However limitations arise in the policies. The toxicity of graphene oxide in the human body is a concept that is currently widely debated, although in some countries, it is already allowed to use these materials. This would influence the idea of ulitizing GO-based composites for water treatment, and then focus on developing methods to remove the GO traces in the water. This is already being explored as well, and we believe it could be the optimal remedy, as the absorption capabilities of GO, and its polymeric composites are crucial in water treatment.

## Figures and Tables

**Figure 1 materials-16-02527-f001:**
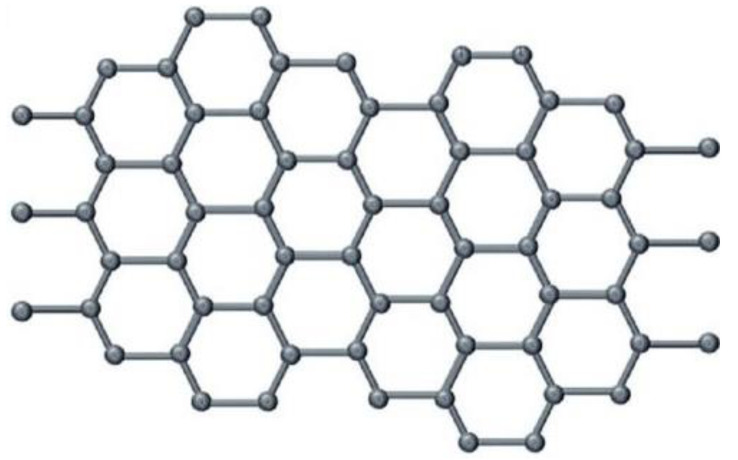
Typical structure of monolayer graphene [44]. (Adapted with permission from ref. [44]. Copyright 2020 IntechOpen.)

**Figure 2 materials-16-02527-f002:**
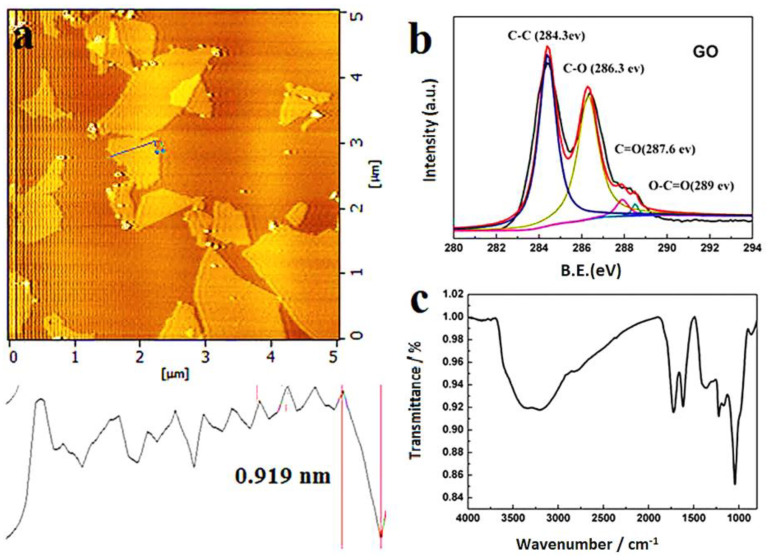
(**a**) AFM image and height profile of GO, (**b**) XPS spectra of GO, and (**c**) FTIR spectra of GO [58]. (Adapted with permission from ref. [58]. Copyright 2016 Taylor and Francis online.)

**Figure 3 materials-16-02527-f003:**
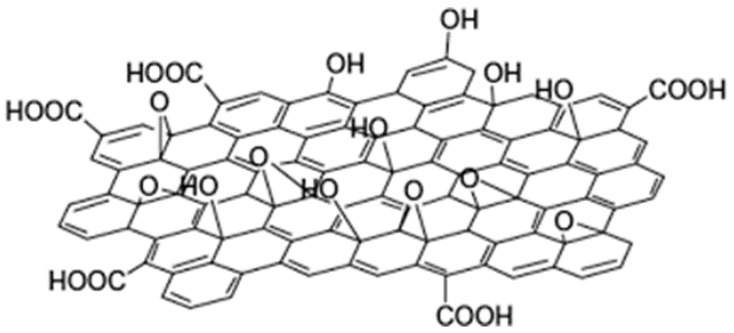
Typical structure of graphene oxide that shows various oxygen-containing functional groups [63]. (Adapted with permission from ref. [63]. Copyright 2015 RCS publishing.)

**Figure 4 materials-16-02527-f004:**
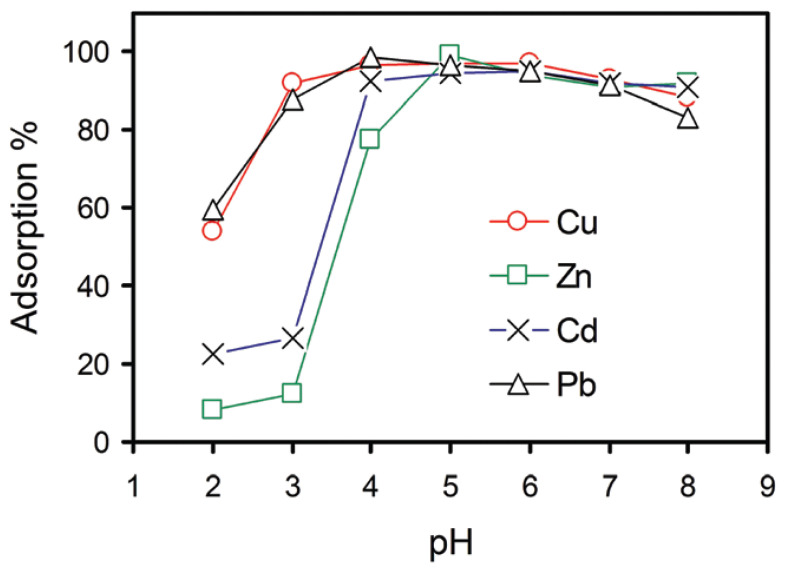
Influence of pH on the adsorption of metal ions on GO. (Adapted with permission from ref. [69]. Copyright 2013 RCS publishing.)

**Figure 5 materials-16-02527-f005:**
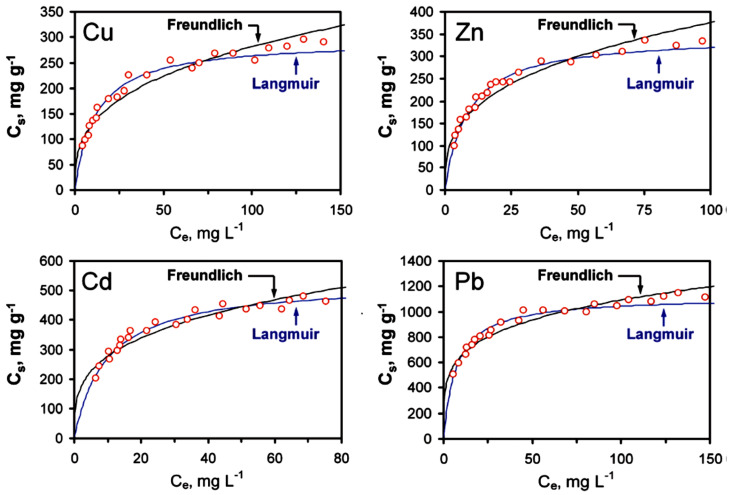
Langmuir and Freundelich isotherms, where the dotted lines represent the individual plots from which the best fit curves were fitted (Adapted with permission from ref. [69]. Copyright 2013 RCS publishing.)

**Figure 6 materials-16-02527-f006:**
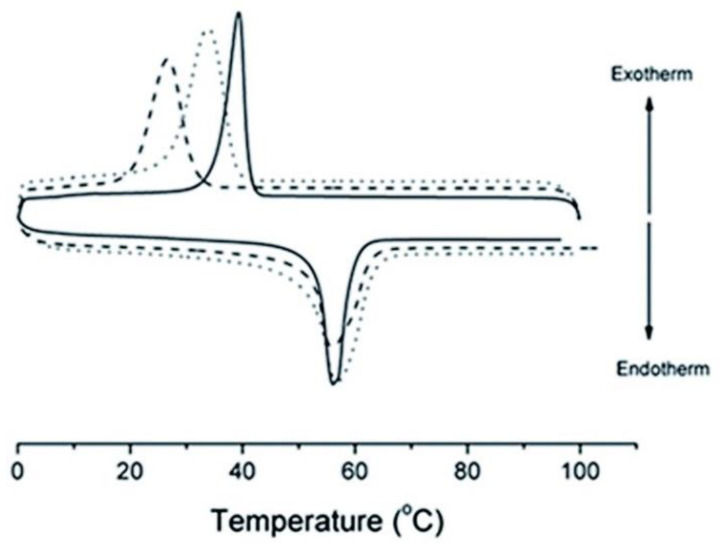
DSC heating and cooling curves for neat PCL (- - -), PCL/rGO (––) composite with 1 wt.% GN loading and PCL/CCG (…) composite with 1 wt.% loading of CCG [73]. (Adapted with permission from ref. [73]. Copyright 2015 RCS publishing.)

**Figure 7 materials-16-02527-f007:**
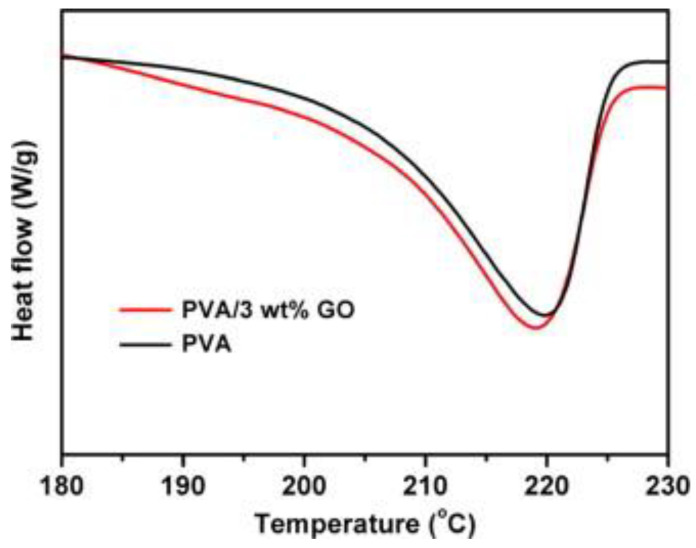
DSC heating curves of PVA/3 wt.% GO composite and neat PVA, endo down [79]. (Adapted with permission from ref. [79]. Copyright 2009 Elsevier.)

**Figure 8 materials-16-02527-f008:**
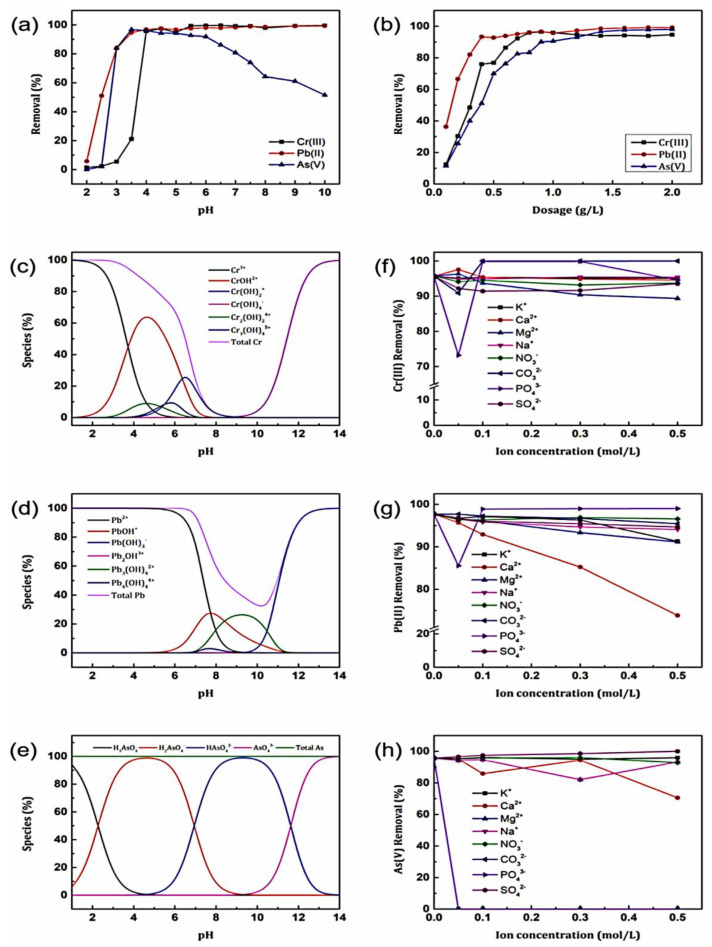
The effect of (**a**) pH and (**b**) dosage of the corresponding metal ions. Graphs of fractions versus ph for: (**c**) Cr(III), (**d**) Pb(II), (**e**) AS(v). Influence of the ion species on the removal of the metal ions: (**f**) Cr(III), (**g**) Pb(II) and (**h**) As(V) [83]. (Adapted with permission from ref. [83]. Copyright 2022 Elsevier.)

**Figure 9 materials-16-02527-f009:**
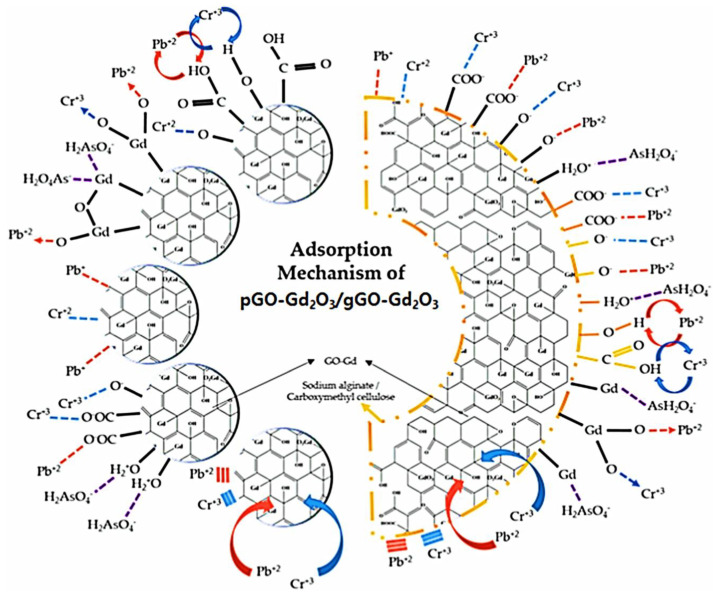
Proposed adsorption mechanism of Cr(III), Pb(II), and As(V) on gGO-Gd_2_O_3_ [83]. (Adapted with permission from ref. [83]. Copyright 2022 Elsevier.)

**Table 1 materials-16-02527-t001:** Weight% elemental compositions of the nickel substrate, carbon, and oxygen under different vegetable oil precursors [46]. (Adapted with permission from ref. [46]. Copyright 2018 AIP publishing.)

Sample		Weight%	
	Nickel	Carbon	Oxygen
Nickel substrate	100	0	0
RCO	82.95	0.96	16.09
RPO	83.12	0.91	15.97
WCPO	77.33	1.54	21.13

**Table 2 materials-16-02527-t002:** The average roughness, Ra, of the graphene growth when different carbon sources are used, and nickel is a substrate [46]. (Adapted with permission from ref. [46]. Copyright 2018 AIP publishing).

Sample	Ra (nm)
Nickel substrate	20.099
Refined corn oil (RCO)	54.794
Refined palm oil (RPO)	61.493
Waste cooking palm oil (WCPO)	69.098

**Table 3 materials-16-02527-t003:** Parameters for Langmuir and Freundlich models of Cu(II), Zn(II), Cd(II), and Pb(II) sorption on GO [69] (Adapted with permission from ref. [69]. Copyright 2013 RCS publishing.)

	Langmuir			Freundlich		
	q_max_ (mg·g^−1^)	K_L_ (L·mg^−1^)	R_L_	K_F_ (mg·g^−1^)(L·mg^−1^)^1/n^	n	R_F_
Cu	294 ± 18	0.09 ± 0.01	0.990	61 ± 8	3.0 ± 0.3	0.974
Zn	345 ± 24	0.12 ± 0.02	0.986	83 ± 11	3.1 ± 0.4	0.965
Cd	530 ± 26	0.10 ± 0.01	0.986	141 ± 20	3.4 ± 0.5	0.955
Pd	1119 ± 41	0.14 ± 0.02	0.983	390 ± 37	4.5 ± 0.5	0.973

**Table 4 materials-16-02527-t004:** DSC data obtained from the second heating scan for pure PLA and its composite with 1 wt.% graphene [72]. (Adapted with permission from ref. [72]. Copyright 2013 ACS publications.)

Sample	T_g_ (°C)	T_cc_ (°C)	ΔH_cc_ (°C)	T_m1_ (°C)	T_m2_ (°C)	X_c_ (°C)
PLA	56.3	101.2	26.25	145.8	154.4	28.44
PLA/G	55.8	103.8	27.28	146.0	154.3	29.23

T_g_—Glass transition temperature, T_cc_—Cold crystallization temperature, ΔH_cc_—Enthalpy of cold crystallization, T_m1_ and T_m2_—Melting points for the first and second melting peaks, respectively.

**Table 5 materials-16-02527-t005:** Water absorption and contact angles of films with various graphene loadings [49]. (Adapted with permission from ref. [49]. Copyright 2011 John Wiley and Sons.)

Sample	Water Absorption (%)	Contact Angle (°)
Pure PVA	105.2	36
0.5 wt.% graphene/PVA	59.8	93
1 wt.% graphene/PVA	48.8	97

**Table 6 materials-16-02527-t006:** Glass transition temperature and melting parameters of neat PVC and PVC/GO composites [81]. (Adapted with permission from ref. [81]. Copyright 2015 Scientific Technical Union of Mechanical Engineering.)

Samples	Glass Transition Temperature (T_g_) (°C)	Melting Temperature (T_m_) (°C)	Enthalpy (ΔH) (J/g)
PVC	59	302	58.66
PVC/GO-0.1	56	299	62.9
PVC/GO-0.3	53	303	57.24
PVC/GO-0.5	54	294	55.41
PVC/GO-1	55	303	29.81

## Data Availability

Not applicable.

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
