# Peer review of "A Review on Graphene (GN) and Graphene Oxide (GO) Based Biodegradable Polymer Composites and Their Usage as Selective Adsorbents for Heavy Metals in Water"

_materials, 2023, doi:10.3390/ma16062527_

Round 1
Reviewer 1 Report
This paper provides a comprehensive review of recent research on graphite, graphene oxide, and their polymer composites from the perspectives of morphology, water absorption, thermal properties, and heavy metal adsorption capabilities. It highlights the promising development prospects of graphene oxide/polymer materials in the field of heavy metal adsorption. The paper is comprehensive, well-organized, and meets the standards for publication in the journal. Nonetheless, there are some limitations, which are elaborated on in detail below.
1. It is suggested to include graphs to demonstrate the heavy metal adsorption capacity of graphene oxide and its polymer composites. This would provide a more intuitive and comprehensive illustration of the feasibility and effectiveness of these materials for heavy metal adsorption treatment
2. It is recommended to summarize the mechanism of heavy metal adsorption to better understand the preparation and regulation goals of graphene oxide/polymer composites, such as increasing the oxygen content, surface area and porosity of the composite materials.
3. To avoid redundancy, it is recommended to refrain from restating information about the main components of thermal analysis in both sections 2.2.2 and 3.1.2.
Author Response
Response to Reviewer 1 Comments
Point 1: It is suggested to include graphs to demonstrate the heavy metal adsorption capacity of graphene oxide and its polymer composites. This would provide a more intuitive and comprehensive illustration of the feasibility and effectiveness of these materials for heavy metal adsorption treatment
Response 1: Graphs have been inserted in the review for heavy metal adsorption capacities of graphene oxide and its polymeric composites. This provided a more intuitive and clearer portrayal of the feasibility and effectiveness of these materials in the removal of heavy metal ions from water.
Point 2: It is recommended to summarize the mechanism of heavy metal adsorption to better understand the preparation and regulation goals of graphene oxide/polymer composites, such as increasing the oxygen content, surface area and porosity of the composite materials.
Response 2: A proposed mechanism has been provided and summarised while reviewing the adsorption studies conducted by reference [83]. This provides clarity regarding the requirements that go into preparing graphene oxide/polymer composites, in order to effectively remove heavy metals from water.
Point 3: To avoid redundancy, it is recommended to refrain from restating information about the main components of thermal analysis in both sections 2.2.2 and 3.1.2.
Response 3: The information regarding the main components of thermal analysis, has been removed from section 3.1.2. It only remains in section 2.2.2, in order to avoid redundancy.
Reviewer 2 Report
The authors have inscribed an extensive review article on graphene and graphene oxide polymeric composites for the removal of heavy metals using adsorption techniques. There are a few comments that need to be revised for the betterment of the manuscript.
1. The authors can focus on references a recent research articles in the field of study, and restrict the references to not less than a decade.
2. In Lines no 50-51 and 54, the references are missing.
3. In fig 2, XP Spectra is not comprehensible, authors can clarify the image.
4. In table 3, the units for the isotherm parameters are missing, it can be represented in the table itself to make readers easy to understand.
5. The authors can highlight the future perspectives, challenges, and policies for GO and GN applications as adsorbents for water treatment.
6. The conclusion part is weaker, the connection between the sentences is not there, especially from 944-952.
Author Response
Response to Reviewer 2 Comments
Point 1: The authors can focus on references a recent research articles in the field of study, and restrict the references to not less than a decade.
Response 1: References have been adjusted to not less than a decade ago for the theory provided in the manuscript. However, where specific authors were outlined, the references were kept as is due to the literature being difficult to find, as this research area has not been explored that much.
Point 2: In lines 50-51 and 54, the references are missing.
Response 2: References have been inserted accordingly in lines 50-51 and 54.
Point 3: In fig 2, XP Spectra is not comprehensible, authors can clarify the image
Response 3: The image in Figure 2 has been clarified in order to make it more comprehensible.
Point 4: In table 3, the units for the isotherm parameters are missing, it can be represented in the table itself to make readers easy to understand.
Response 4: Isotherm parameter units have been added accordingly and where fitting in table 3, in order to help readers understand.
Point 5: The authors can highlight the future perspectives, challenges , and policies for GO and GN applications as adsorbents for water treatment.
Response 5: The challenges, future perspectives and general policies for GO and GN applications as adsorbents in water, have been highlighted under the ’conclusions and future perspectives’ section.
Point 6: The conclusion part is weaker, the connection between sentences is not there, especially from 944-952.
Response 6: The conclusion has been revised in order to achieve clarity and a connection in between the sentences.